# Child and caregiver mental health during 12 months of the COVID-19 pandemic in Australia: findings from national repeated cross-sectional surveys

Anna MH Price ![ORCID] [1,2,3] Mary-Anne Measey,[4] Monsurul Hoq,[5] Anthea Rhodes,[4] Sharon Goldfeld[1,2,3]

AR and SG are joint senior authors.

For numbered affiliations see end of article.

**Correspondence to**
Dr Anna MH Price; anna.price@mcri.edu.au

## ABSTRACT

**Background** There are calls for research into the mental health consequences of living through the COVID-19 pandemic. Australia's initial, effective suppression of COVID-19 offers insights into these indirect impacts in the relative absence of the disease. We aimed to describe the mental health experiences of Australian caregivers and children over 12 months, reporting differences related to demographic, socioeconomic and lockdown characteristics.

**Methods** Data were from Australia's only nationally representative, repeated cross-sectional survey of caregivers with children (0–17 years). N=2020 caregivers participated in June 2020, N=1434 in September 2020 and N=2508 in July 2021. Caregivers reported their mental health (poor vs not, Kessler-6), and perceived impacts of the pandemic on theirs and their children's mental health (negative vs none/positive). Data were weighted to approximate population distributions of caregiver age, gender, sole caregiving, number and ages of children, state/territory and neighbourhood-level disadvantage.

**Results** Perceived impacts on mental health were more frequently negative for female (vs male) caregivers and older (vs younger) children. Poor caregiver mental health (Kessler-6) was more common for families experiencing socioeconomic adversity (especially financial), while perceived impacts were more frequently negative for more socially advantaged groups. Caregivers who experienced the least total lockdown reported similar mental health over time. Otherwise, poor mental health and perceived negative impacts increased over time with increasing total length of lockdown.

**Conclusion** Despite Australia's low infection rates, the negative mental health experiences of the COVID-19 pandemic are real and concerning. Addressing poor mental health must be central to ongoing pandemic recovery efforts for families and children.

## WHAT IS ALREADY KNOWN ON THIS TOPIC

⇒ The global evidence shows that, for general adult populations, psychological distress peaked in the first months of the COVID-19 pandemic before appearing to improve.
⇒ Less is known about the longer-term mental health experiences of living through the pandemic, especially for caregivers and children. There are urgent calls for research.
⇒ Due to low infection rates, Australia's experience can provide insight into the mental health impacts of lockdown with minimal compounding harms of the virus.

## WHAT THIS STUDY ADDS

⇒ From June 2020 to July 2021, Australia's lockdowns were detrimental for caregiver and child mental health.
⇒ Negative mental health experiences differed by caregiver gender, child age and family socioeconomic characteristics.
⇒ Pandemic response and recovery planning must consider both family mental health and socioeconomic security.

## HOW THIS STUDY MIGHT AFFECT RESEARCH, PRACTICE OR POLICY

⇒ Mental health is changing during the pandemic. Understanding families' experiences is necessary to inform policy effort with a greater level of precision, so that we can respond to the evolving mental health needs of children and their families.

## INTRODUCTION

During the first 18 months of the COVID-19 pandemic, balancing the direct harms of the virus with the indirect harms of public health restrictions was a global challenge. Australia enacted some of the strictest public health measures in the world to effectively suppress infection.[1] By 31 July 2021, the measures successfully contained infection to an overall incidence rate of 137 cases and 3.7 deaths per 100 000 people.[2] With just 15% of the population of 25 million fully vaccinated by the end of July 2021, stay-at-home orders ('lockdown') were the country's main measure of disease control.[3] Families caring for children and young people (referred to as children hereafter) were substantially affected by lockdown,[4 5] required to juggle working from home, responsibility for

their children's remote learning and care during school and childcare closures, with minimal social opportunities due to closure of sports and play centres, and reduced access to healthcare. Despite mental health having a 'greater impact on human activity than any other non-communicable illness',[4] it appeared to be given little consideration in the majority of Australia's lockdown policies. With future variants and pandemics likely, Australia's experience offers insight into the mental health experiences of lockdown, effectively without the compounding direct harms of the virus.

The evidence for the indirect and longer-term impacts of the pandemic is quickly evolving. In a review of the global mental health evidence from the first year (to April 2021), Aknin *et al*, reported a peak in adults' psychological distress in the early months.[4] While many studies reported a decline to pre-pandemic levels by mid-2020,[4] the authors found that mental health inequities were sustained or exacerbated for adults who were younger, female, child-rearing or with fewer socioeconomic resources.[4 5] However, published data on the specific experience of parents (termed caregivers throughout) are scarce. Evidence reviews of children's experiences also suggest that their mental health declined during the pandemic.[6 7] In Racine *et al*'s meta-analysis of 29 studies published in the first year of the pandemic, prevalence estimates for depression and anxiety in children doubled, and were higher over time, and for older adolescents and females.[8] However, the available systematic reviews are limited by over-representation of data from the early months of the pandemic,[6–8] and previous pandemics show that negative mental health effects can persist.[9] As such, there are urgent calls for research into the 'long-term mental health consequences of living through the pandemic',[4] which prioritises children.[5]

In response to these calls, we aimed to describe the mental health experiences of caregivers and children during the pandemic, using Australia's only nationally representative, repeated cross-sectional survey of families with children. Unlike many high-income countries, Australia's low incidence of COVID-19 over the first 18 months of the pandemic makes it possible to examine the mental health experiences in the relative absence of the disease. Based on the gender, age and socioeconomic disparities identified in the published evidence reviews,[4 8] we hypothesised that caregiver mental health and perceived impacts of the pandemic on caregiver and child mental health would be worse: (1) for caregivers who identified as female compared with male, and older compared with younger children, (2) for families experiencing greater socioeconomic adversity compared with more socioeconomic advantage and (3) with increasing total length of lockdown.

## METHODS
### Design and procedure
The Royal Children's Hospital (RCH) National Child Health Poll comprises periodic cross-sectional surveys of approximately 2000 Australian caregivers of children aged 0–17 years. Data collection is contracted to the Online Research Unit that obtain written informed consent and draw a nationally representative sample of caregivers using stratified random sampling from their panel of over 350 000 adults aged 18 years or older, who live in Australia and have internet access. Surveys are administered in English, with a reading level equivalent to sixth grade (the end of elementary school). Responses are anonymous, and respondents are remunerated with points exchangeable for department store gift vouchers.

Questions about mental health—the focus of the current study—were introduced after COVID-19 began and were collected in three surveys. The dates of data collection were: (1) 15–23 June 2020, after a first national lockdown (March–May 2020) eased; (2) 15–29 September 2020, when only metropolitan residents of Victoria were in a second, stricter lockdown (July–November 2020); and (3) 20–29 July 2021, when multiple states/territories were in and out of lockdown (June 2021 onwards).

### Patient and public involvement
The research questions and design of this study were informed by previous RCH Poll surveys, which asked caregivers to identify the child health issues of most concern to them and which child health topics should be included in future polls. At the end of each survey, participants were informed of the study website where all research reports are accessible to the public. As each survey is collected from a cross-sectional, population-based online survey of a random sample, respondents were not directly involved in the recruitment or conduct of the study.

### Measures
Each survey collected the demographic, socioeconomic and pandemic-related financial characteristics described in table 1. To achieve high response rates and population representativeness, the surveys are intentionally brief and ask questions that are simple in their phrasing and response options. Table 1 also describes how the two measures of lockdown (current and total length) were defined according to Australia's state and federal government pandemic responses. Families were assigned the Australian Bureau of Statistics' (ABS) Socio-Economic Indexes for Areas (SEIFA) Index of Relative Disadvantage, a national area level index derived from census data for all individuals living in a postcode, with higher scores indicating greater advantage. Caregivers self-reported their mental health (Kessler-6 (K6)), and the perceived impact of the pandemic on their own mental health, and the mental health of each child in their care (details in table 1).

### Analysis
To reduce effects of non-response and non-coverage and therefore approximate the population distributions of mental health experiences, the mental health measures were weighted using ABS distributions of caregiver age,

**Table 1** Demographic, socioeconomic, lockdown and mental health measures

| Measure | Description |
|---|---|
| **Demographic** | |
| Age | Collected for caregivers and children, reported in years. Families and children were differentially affected by lockdown depending on whether children were attending early education centres (day care) or school. This is because schools and early education centres were closed for different, and varying periods of the pandemic. Caregivers of school aged children were required to facilitate home learning, which was arguably more difficult for caregivers of young (compared with older) children. Similarly, families were required to care for young children when early education centres closed, which made it difficult to work from home and also balance other responsibilities. Child age was used as a proxy for these experiences, and categorised to represent preschool (0–4 years), primary/elementary school (5–11 years) and high school (12–17 years). |
| Gender | Collected for caregivers and children: response options 'male', 'female', 'other'. Two caregivers and no children identified as 'other'. Before the 2020 Australian census, 'other' gender data were not collected for the Australian population, so it was not possible to calculate weights for these caregivers. Hence, they were dropped from the analysis, and the gender variable was dichotomised into 'female' compared with 'male'. |
| Sole caregiver | Question 'are you the sole (single) parent or carer of a child 17 years of age or younger?', binary response options 'yes' (one caregiver household) compared with 'no' (multicaregiver household). |
| Caregiver education | Question 'what is the highest level of schooling/education you have completed?'. Responses were trichotomised into three categories that meaningfully represented education as a socioeconomic measure for Australians: (1) 'year 12 or less' (response options: less than year 10, year 10 or equivalent (eg, school certificate), year 12 or equivalent); (2) 'vocational training certificate' (response options: trade/apprenticeship (eg, carpenter), certificate/diploma (eg, Cert IV Childcare)); or (3) 'university degree' (response options: undergraduate university degree, postgraduate university degree (eg, Masters, Doctorate, PhD). |
| Caregiver country of birth | Question 'were you born in Australia?', binary response options 'yes' (born in Australia) compared with 'no' (outside Australia). |
| Home language | Question 'do you speak a language other than English at home?', binary response options 'yes' (other than English) compared with 'no' (English). |
| Cultural or linguistic diversity | A composite of the two above variables, caregiver country of birth and home language, to represent respondents who answered 'yes' to either or both items compared with 'no' (answering no to both items). |
| **Socioeconomic and lockdown** | |
| Job/income loss during COVID-19 | Three items drawn from the CoRonavIruS Health Impact Survey caregiver version[13]: 'what changes in employment or income have occurred in your household due to coronavirus/COVID-19?' (binary response options 'yes' compared with 'no') including: 'job loss by one caregiver'; 'job loss by two caregivers' and 'reduced total household income'. A binary variable describing any job loss (by one or two caregivers) or reduction in income due to COVID-19 (compared with not) was created. |
| Low income (<$A1000) | A binary variable based on current total household income before tax, categorised into 10 options ranging from 'less than $A500 per week' to 'more than $A3000 per week', plus 'prefer not to say'. In 2021, to protect against the economic fallout of lockdown, the Australian federal government rapidly implemented a suite of short-term financial supports, which included an unemployment supplement ('JobSeeker') which doubled recipients' social welfare benefits from $A550 to $A1100 a fortnight, and a wage supplement for eligible businesses to retain their workforce ('JobKeeper'). These social policy changes represented some of the largest (although temporary) in Australia's history and were so significant that, by September 2020, levels of poverty and housing stress in Australia were substantially lower than the levels directly preceding COVID-19. To capture any relationships between income poverty and mental health, we created a binary variable summarising low income ('less than $A1000 per week' compared with more) based on Australian thresholds.[14] n=767 caregivers preferred not to report income, so this variable should be interpreted with caution. |
| Could not afford essential items | Eight items adapted from the Household, Income and Labour Dynamics in Australia Survey Wave 18 Household Questionnaire Material Deprivation Module[15] asking 'In the last month, because of money pressure did you miss or put off' (binary response options: 'yes' compared with 'no'): mortgage or rent repayments; electricity, gas, water bills; food; healthcare; prescription medicines; home or car insurance; mobile phone bills; and internet. A binary summary variable was created to denote 'any material deprivation' inability to pay for one or more essential items compared with 'none'. |
| Current lockdown | All Australian states and territories experienced a first national, 10-week lockdown from 23 March to 1 June 2020. Victoria experienced a second extended and more severe, 20-week lockdown from 8 July to 23 November 2020. There were several short (less than a week) lockdowns in the intervening months across states, before New South Wales experienced its second extended period of lockdown, which began incrementally on 26 June 2021. Victoria experienced its fifth (in total) lockdown from 16 to 27 July 2021, before an extended lockdown subsequently began a week later on 5 August 2021. Current lockdown was therefore categorised as follows.<br>▶ The national lockdown had eased for the collection of the June 2020 poll, so no (denoted 'N' in tables 2 and 3) Australians were in lockdown. Thus, the number/proportion not in lockdown represents the whole cohort (and is equal to the 'overall' numbers/proportions at top of tables 2 and 3).<br>▶ For the September 2020 poll, only residents of metropolitan Melbourne in the state of Victoria were in lockdown (denoted 'Y' for 'yes') compared with all other Australians (N). This was categorised according to respondents who reported living in Victoria and living in 'metropolitan' compared with 'regional/rural' areas.<br>▶ By July 2021, many states/territories were going in and out of lockdown, and this question was added into the poll and reported on directly with the question 'are you currently under stay-at-home orders or restrictions due to COVID-19 (also known as 'lockdown')?', responses 'yes' compared with 'no'. |
| Total length of lockdown | Trichotomous variable based on total length of lockdown experienced by each state/territory. By 31 July 2021, the total length of COVID-19 lockdown was greatest for the state of Victoria ('Vic', total 31 weeks); followed by the state of New South Wales ('NSW', total 15 weeks) and then all 'other' states and territories (total range 10–12 weeks). The following geographical categories were used as a proxy for total length of lockdown: (1) Victorian (most), (2) NSW and (3) other (least). |
| **Mental health** | |
| Caregiver mental health | 6 items of the Kessler-6 (K6) assessing caregivers' self-reported anxiety and depressive symptoms encountered in the last 4 weeks. Scored on a 5-point Likert scale from 1 'none of the time' to 5 'all of the time'. Summarised into (1) a continuous total score and (2) a binary variable indicating 'poor mental health' (total score 19 or more) compared with not (total score 6–18).[16] |
| Perceived impact of the pandemic on mental health | A 5-point item adapted from UK Young Minds Matter Study,[17] describing the perceived impact of COVID-19 on mental health, dichotomised into negative ('small negative/large negative') compared with positive ('none/small positive/large positive'). Reported by caregivers for (1) themselves and (2) each child. |

gender, family structure (sole caregiving, number of children and any under 5 years), state/territory and SEIFA. As the primary aim of this study was to describe the mental health experiences of Australian families during the pandemic, we calculated simple weighted proportions of the mental health measures for each survey. These were described overall and by demographic, socioeconomic and pandemic-related financial and lockdown characteristics.

To investigate the specific relationship between lockdown and mental health, we used generalised linear models to calculate the association between each mental health measure and time of the survey, according to state/territory grouping (as a proxy for total length of lockdown, see table 1). The binomial models investigated whether increasing length of lockdown increased the frequency of negative mental health experiences over time, after adjusting for caregiver gender, sole caregiver status, education, SEIFA and cultural or linguistic diversity (measures described in table 1). The child mental health model was additionally adjusted for poor caregiver mental health (K6, see table 1) and clustering at the level of family. The estimated models were then used to predict probabilities of each mental health outcome, by state/territory grouping, at the three survey timepoints, adjusted for the covariates in the model.

## RESULTS

In June 2020, n=2020 of 2697 (74.9%) caregivers approached provided data for themselves and 3411 children. In September 2020, 1434/1769 (81.1%) caregivers provided data for 2553 children. In July 2021, 2508/3925 (63.9%) provided data for 4327 children. Table 2 presents the sample characteristics for each survey. The SEIFA quintiles suggested strong response bias towards more advantaged groups. There were also some differences between surveys in characteristics such as the proportion of respondents caring for young children, caregiver gender, sole caregiving and SEIFA; characteristics that were used to create the sample weights (see Analysis).

For the three surveys overall, children's mean age was 9.3 years (SD 5.0 years) and 47.8% were girls. Caregivers' mean age was 42.2 years (SD 9.1 years), 54.0% were women, and each caregiver cared for 1.7 children on average (SD 0.8). One-quarter were sole caregivers, 56.6% had a university degree, 25.6% were born outside Australia and 22.4% spoke a language other than English at home. In total, 28% of caregivers reported job/income loss due to COVID-19, 17.2% reported low household income and 29.3% were unable to afford at least one essential item in the month prior. The proportions of respondents in lockdown were 0% (June 2020), 29.2% (September 2020) and 56.5% (July 2021).

Table 3 presents the weighted mental health experiences for each survey, separately by characteristics. When considering patterns over time, the proportions of respondents who reported poor mental health according

to the K6 was similar between surveys. Perceived impacts on caregiver mental health became more frequently negative between June 2020 and July 2021, although there were no consistent patterns between June 2020 and September 2020, or between September 2020 and July 2021. In contrast, there was a clear pattern that perceived impacts on children's mental health became more frequently negative across the three waves of data collection.

When considering differences between subgroups, perceived negative impacts on mental health were more common for caregivers and children when reported by female than male caregivers, and for older compared with younger children, but were similar between child genders. Poor mental health (K6) was more common for groups typically experiencing more social adversity, such as sole caregiving compared with multicaregiver households, having a home language other than English (but not for country of birth), and several financial and lockdown characteristics (lower SEIFA and income, could not afford essentials, and increasing total or current lockdown). In contrast with the K6, perceived impacts of the pandemic on caregiver and child mental health were more frequently negative for families experiencing greater social advantage, such as multicaregiver (compared with sole caregiver) households and higher (compared with lower) education, SEIFA and income. Job or income loss, and increasing total length of lockdown, were associated with increasing negative mental health experiences across all three measures, across the three timepoints.

Figure 1 shows that the three measures of negative mental health experiences increased for Victorian and New South Wales (NSW) families by July 2021 compared with June 2020. Increases in the proportions of caregivers reporting poor mental health (K6) aligned with the second extended lockdowns; for Victorians this was September 2020 and for NSW it was July 2021 (figure 1A, online supplemental table 1A). In contrast, there was weak evidence of change in the proportions of caregivers in other states/territories reporting poor mental health (K6) over time. The proportions of caregivers and children who perceived negative impacts of the pandemic on their mental health increased from June 2020 to July 2021 (figure 1B,C, online supplemental table 1B,C). Increased proportions of Victorians perceived negative impacts on their mental health in September 2020 and again in July 2021. More caregivers from NSW and the other states/territories self-reported negative impacts in July 2021 compared with June or September 2020.

## DISCUSSION

This study describes the mental health experiences captured by the only nationally representative and repeated cross-sectional survey of Australian families over 12 months of the COVID-19 pandemic. Poor caregiver mental health (K6) was stable and similar between male

**Table 2** Sample characteristics by survey, described with the number of respondents (proportion)

| Characteristic | Subgroup | Respondents, n (%) | | |
| --- | --- | --- | --- | --- |
| | | June 2020 | September 2020 | July 2021 |
| **Child** | | **N=3411** | **N=2553** | **N=4327** |
| Age in years | 0–4 | 628 (18.4) | 520 (20.4) | 1291 (29.8) |
| | 5-11 | 1356 (39.8) | 1153 (45.2) | 1705 (39.4) |
| | 12-17 | 1427 (41.8) | 880 (34.5) | 1331 (30.8) |
| Gender | F | 1664 (48.7) | 1242 (48.7) | 2013 (46.5) |
| | M | 1747 (51.2) | 1311 (51.4) | 2314 (53.5) |
| **Caregiver** | | **N=2020** | **N=1434** | **N=2508** |
| Gender | F | 1003 (49.7) | 758 (52.9) | 1458 (58.1) |
| | M | 1017 (50.4) | 676 (47.1) | 1050 (41.9) |
| Sole carer | Yes | 462 (22.9) | 282 (19.7) | 704 (28.1) |
| | No | 1558 (77.1) | 1152 (80.3) | 1804 (71.9) |
| Education* | Y12 | 326 (16.1) | 239 (16.7) | 422 (16.8) |
| | Cert. | 518 (25.6) | 390 (27.2) | 693 (27.6) |
| | Uni. | 1176 (58.2) | 805 (56.1) | 1393 (55.5) |
| Born outside Australia | Yes | 546 (27.4) | 329 (23.4) | 622 (25.4) |
| | No | 1444 (72.6) | 1078 (76.6) | 1827 (74.6) |
| Home language other than English | Yes | 413 (20.5) | 324 (22.6) | 600 (23.9) |
| | No | 1607 (79.6) | 1110 (77.4) | 1908 (76.1) |
| Socioeconomic | | | | |
| SEIFA quintile (1=most disadvantage, 5=least disadvantage) | 1 | 213 (10.6) | 172 (12.0) | 327 (13.0) |
| | 2 | 273 (13.5) | 245 (17.1) | 391 (15.6) |
| | 3 | 370 (18.3) | 250 (17.4) | 511 (20.4) |
| | 4 | 457 (22.7) | 319 (22.3) | 571 (22.8) |
| | 5 | 705 (34.9) | 448 (31.2) | 707 (28.2) |
| Low income (<$A1000)† | Yes | 254 (14.7) | 212 (16.9) | 427 (19.2) |
| | No | 1480 (85.3) | 1040 (83.1) | 1782 (80.7) |
| Could not afford essential items‡ | Yes | 527 (26.1) | 400 (27.9) | 822 (32.8) |
| | No | 1493 (73.9) | 1034 (72.1) | 1686 (67.2) |
| Pandemic experience | | | | |
| Job/income loss§ | Yes | 559 (27.7) | 381 (26.6) | 711 (28.4) |
| | No | 1461 (72.3) | 1053 (73.4) | 1797 (71.7) |
| Current lockdown¶ | Yes | 0 (0) | 419 (29.2) | 1416 (56.5) |
| | No | 2020 (100) | 1015 (70.8) | 1092 (43.5) |
| State as a proxy for total length of lockdown (Vic=most, other=least)¶ | Vic | 612 (30.3) | 473 (33.0) | 746 (29.7) |
| | NSW | 617 (30.5) | 449 (31.3) | 791 (31.5) |
| | Other | 791 (39.2) | 512 (35.7) | 971 (38.7) |

Data were weighted using national demographic distributions for caregiver age, gender, family structure (sole caregiving, number of children and any under 5 years), state/territory and Socio-Economic Indexes for Areas Index of Relative Disadvantage (SEIFA).
*Highest education coded as year 12 or less (up to the end of high/secondary school; Y12); vocational training certificate ('cert.'); or university degree ('uni').
†Low income defined according to Australian definitions of income poverty. Missing 767 caregivers who preferred not to report income.
‡Any one or more of mortgage or rent; electricity, gas, water bills; food; healthcare; prescription medicines; home or car insurance; mobile phone bills; internet, in the last month.
§Job loss by one or two adults, or reduction in income, due to COVID-19.
¶No Australians were in lockdown for the June 2020 poll, so the number/proportion is for the whole cohort. For the September 2020 poll, only residents of metropolitan Melbourne in the state of Victoria were in lockdown (Y) compared with all other Australians (N). In July 2021, many states/territories were going in and out of lockdown, and this question was asked directly (Y/N). Overall, Victorians experienced the longest total lockdown, followed by NSW, followed by other states and territories.
NSW, New South Wales; Vic, Victoria.

**Table 3** Mental health experiences by sample characteristic and survey, described with the number of respondents (weighted proportion)

| Characteristic | Subgroup | Poor caregiver mental health (K6)* | | | Perceived negative impact on caregiver mental health† | | | Perceived negative impact on child mental health† | | |
|---|---|---|---|---|---|---|---|---|---|---|
| | | June 2020 N=2020 | September 2020 N=1434 | July 2021 N=2508 | June 2020 N=2020 | September 2020 N=1434 | July 2021 N=2508 | June 2020 N=3411 | September 2020 N=2553 | July 2021 N=4327 |
| Overall | All | 264 (17.4) | 247 (19.5) | 541 (18.9) | 949 (47.4) | 708 (50.3) | 1490 (60.0) | 1055 (25.6) | 884 (33.1) | 1807 (44.0) |
| Child | | | | | | | | | | |
| Age in years | 0–4 | – | – | – | – | – | – | 81 (12.8) | 101 (19.9) | 330 (24.2) |
| | 5–11 | – | – | – | – | – | – | 440 (32.3) | 414 (37.3) | 789 (46.9) |
| | 12–17 | – | – | – | – | – | – | 534 (38.2) | 369 (43.7) | 688 (51.5) |
| Gender | F | – | – | – | – | – | – | 525 (25.1) | 438 (32.8) | 876 (45.5) |
| | M | – | – | – | – | – | – | 530 (26.0) | 446 (33.4) | 931 (42.7) |
| Caregiver | | | | | | | | | | |
| Gender | F | 153 (18.3) | 110 (18.0) | 338 (20.5) | 489 (49.6) | 410 (53.9) | 920 (63.7) | 512 (25.9) | 521 (35.1) | 1099 (45.7) |
| | M | 111 (15.8) | 137 (21.8) | 203 (16.6) | 460 (44.1) | 298 (44.7) | 570 (54.7) | 543 (25.1) | 363 (30.0) | 708 (41.6) |
| Sole carer | Yes | 137 (36.1) | 100 (36.5) | 268 (32.6) | 199 (42.7) | 99 (36.2) | 335 (44.3) | 190 (24.3) | 144 (27.1) | 397 (30.9) |
| | No | 127 (10.2) | 147 (14.8) | 273 (15.0) | 750 (49.2) | 609 (54.1) | 1155 (64.5) | 865 (26.0) | 740 (34.8) | 1410 (47.9) |
| Education‡ | Y12 | 55 (21.7) | 61 (26.9) | 124 (15.8) | 141 (40.2) | 98 (42.9) | 211 (47.3) | 148 (21.2) | 115 (24.3) | 293 (39.2) |
| | Cert. | 73 (21.4) | 65 (19.9) | 152 (20.0) | 244 (48.7) | 204 (56.1) | 418 (64.6) | 293 (28.6) | 264 (34.9) | 524 (51.1) |
| | Uni. | 136 (15.8) | 121 (16.9) | 265 (17.5) | 564 (48.8) | 406 (49.6) | 861 (61.5) | 614 (25.3) | 505 (34.9) | 990 (41.8) |
| Born outside Australia | Yes | 58 (12.5) | 40 (11.7) | 124 (16.9) | 243 (42.4) | 169 (49.2) | 379 (60.0) | 253 (23.0) | 184 (32.0) | 396 (40.2) |
| | No | 198 (18.9) | 198 (21.0) | 405 (19.6) | 694 (49.4) | 530 (50.8) | 1080 (60.1) | 789 (26.8) | 695 (33.7) | 1375 (45.3) |
| Home language other than English | Yes | 80 (20.9) | 79 (22.5) | 176 (25.1) | 172 (40.6) | 154 (47.6) | 337 (58.9) | 178 (21.7) | 181 (31.7) | 352 (39.5) |
| | No | 184 (16.3) | 168 (18.6) | 365 (17.0) | 777 (49.4) | 554 (51.1) | 1153 (58.9) | 877 (26.8) | 703 (33.5) | 1455 (45.4) |
| Socioeconomic | | | | | | | | | | |
| SEIFA quintile (1=most disadvantage, 5=least disadvantage) | 1 | 41 (22.1) | 35 (20.7) | 97 (26.1) | 86 (41.4) | 77 (47.4) | 172 (56.0) | 115 (27.1) | 90 (31.3) | 223 (42.0) |
| | 2 | 50 (20.5) | 53 (25.6) | 99 (21.4) | 125 (46.1) | 117 (51.3) | 215 (60.6) | 133 (21.1) | 157 (32.6) | 262 (45.1) |
| | 3 | 49 (16.2) | 41 (19.3) | 114 (17.9) | 186 (53.1) | 129 (51.9) | 295 (58.7) | 191 (27.3) | 170 (34.5) | 350 (40.8) |
| | 4 | 66 (17.9) | 54 (20.8) | 108 (16.0) | 205 (44.1) | 149 (49.6) | 350 (62.4) | 242 (26.6) | 172 (31.0) | 392 (42.8) |
| | 5 | 58 (9.4) | 64 (11.1) | 123 (16.5) | 347 (51.7) | 236 (51.0) | 458 (67.1) | 374 (26.3) | 295 (35.7) | 580 (52.9) |
| Low income (<$A1000)§ | Yes | 69 (27.5) | 58 (31.3) | 171 (29.5) | 120 (53.4) | 89 (45.6) | 230 (51.2) | 111 (26.1) | 133 (33.3) | 271 (35.3) |
| | No | 165 (15.5) | 167 (17.3) | 326 (16.6) | 686 (45.2) | 507 (49.2) | 1064 (61.0) | 774 (24.8) | 631 (32.4) | 1292 (44.6) |
| Could not afford essential items¶ | Yes | 359 (37.4) | 156 (38.4) | 341 (35.5) | 263 (50.3) | 165 (44.8) | 469 (59.2) | 270 (26.5) | 236 (33.9) | 586 (42.6) |
| | No | 96 (8.8) | 91 (12.3) | 200 (11.2) | 686 (46.2) | 543 (52.4) | 1021 (60.4) | 785 (25.1) | 648 (32.8) | 1221 (44.7) |
| Pandemic experience | | | | | | | | | | |
| Job/income loss** | Yes | 96 (23.7) | 93 (27.5) | 218 (28.8) | 306 (51.0) | 224 (59.5) | 486 (71.1) | 305 (25.0) | 297 (40.7) | 613 (55.8) |
| | No | 168 (15.1) | 288 (16.4) | 323 (15.3) | 643 (46.1) | 484 (46.8) | 1004 (55.9) | 750 (25.7) | 587 (30.2) | 1194 (39.7) |

Continued

**Table 3** Continued

| Characteristic | Subgroup | Poor caregiver mental health (K6)* | | | Perceived negative impact on caregiver mental health† | | | Perceived negative impact on child mental health† | | |
|---|---|---|---|---|---|---|---|---|---|---|
| | | June 2020 N=2020 | September 2020 N=1434 | July 2021 N=2508 | June 2020 N=2020 | September 2020 N=1434 | July 2021 N=2508 | June 2020 N=3411 | September 2020 N=2553 | July 2021 N=4327 |
| Current lockdown†† | Yes | n/a | 83 (24.0) | 353 (23.5) | n/a | 240 (58.1) | 867 (62.9) | n/a | 355 (44.1) | 1114 (49.6) |
| | No | 264 (17.4) | 164 (17.7) | 188 (13.3) | 949 (47.4) | 468 (47.3) | 623 (56.4) | 1055 (25.6) | 529 (28.7) | 693 (37.3) |
| State as a proxy for total length of lockdown (Vic=most, other=least)†† | Vic | 85 (17.7) | 93 (23.3) | 200 (25.1) | 284 (48.9) | 266 (57.5) | 483 (66.4) | 321 (23.8) | 389 (42.5) | 626 (51.4) |
| | NSW | 96 (21.6) | 78 (18.6) | 188 (22.0) | 307 (49.2) | 213 (49.0) | 477 (60.8) | 306 (25.9) | 267 (34.4) | 603 (48.9) |
| | Other | 83 (12.4) | 76 (16.6) | 153 (11.2) | 358 (43.9) | 229 (44.1) | 530 (54.4) | 428 (26.9) | 228 (21.0) | 578 (33.7) |

Proportions were weighted using national demographic distributions for caregiver age, gender, family structure (sole caregiving, number of children and any under 5 years), state/territory and Socio-Economic Indexes for Areas Index of Relative Disadvantage (SEIFA).

*Kessler-6 (K6) dichotomised into a binary 'poor mental health' (total score 19 or more) compared with not (total score 6–18).

†Dichotomised into negative ('small negative/large negative') compared with positive ('none/small positive/large positive').

‡Highest education coded as year 12 or less (up to the end of high/secondary school; Y12); vocational training certificate ('cert.'); or university degree ('uni').

§Low income defined according to Australian definitions of income poverty. Missing 767 caregivers who preferred not to report income.

¶Any one or more of mortgage or rent; electricity, gas, water bills; food; healthcare; prescription medicines; home or car insurance; mobile phone bills; internet, in the last month.

**Job loss by one or two adults, or reduction in income, due to COVID-19.

††No Australians were in lockdown for the June 2020 poll, so the proportion is for the whole cohort (see 'overall' at top of table). For the September 2020 poll, only residents of metropolitan Melbourne in the state of Victoria were in lockdown (Y) compared with all other Australians (N). In July 2021, many states/territories were going in and out of lockdown, and this question was asked directly (Y/N). Overall, Victorians experienced the longest total lockdown, followed by NSW, followed by other states and territories.

NSW, New South Wales; Vic, Victoria.

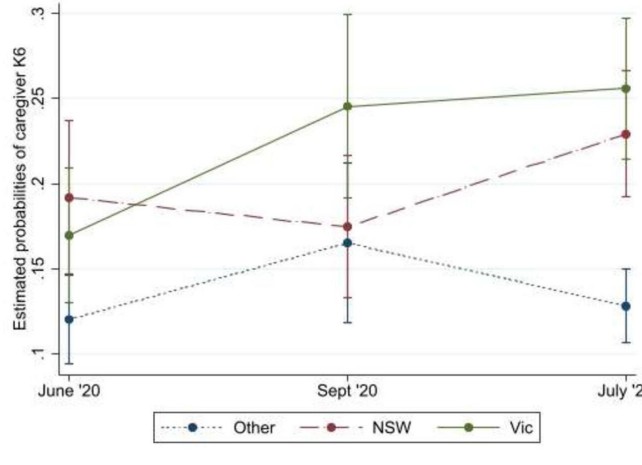

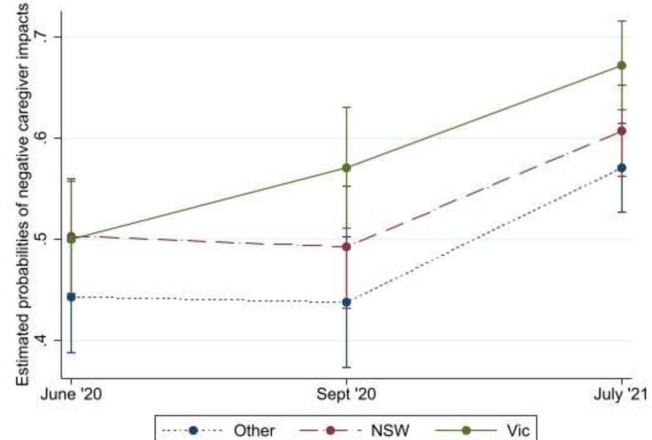

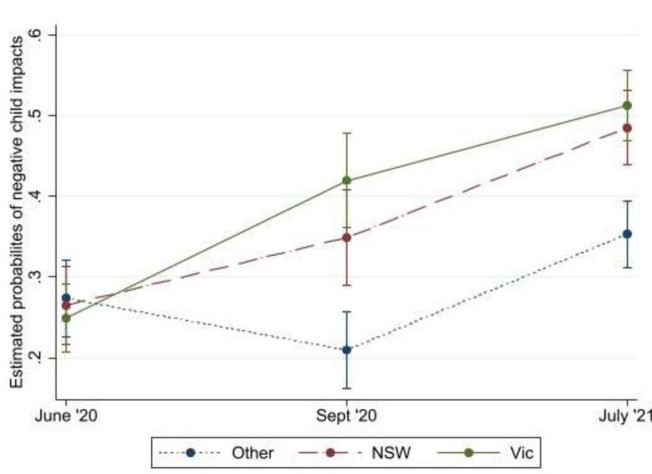

**Figure 1** Estimated probabilities over time (by survey) and by state/territory (as a proxy for total length of lockdown) adjusted for potential confounders for (A: top Figure) caregiver K6, and negative mental health impacts for the (B: middle Figure) caregiver and (C: bottom Figure) children. NSW, New South Wales; Vic, Victoria.

and female genders. In contrast, the proportion of caregivers reporting a negative impact on mental health increased over time, and was more common for female caregivers, and older (compared with younger) children. Across the three surveys, job or income loss due to the

pandemic was associated with poorer caregiver mental health and perceived negative impacts on caregiver and child mental health. While poor caregiver mental health was more common for families experiencing greater social adversity, perceived impacts on mental health were more frequently negative for the more socially advantaged groups. Caregivers in states/territories who experienced the least total lockdown reported similar K6 scores over time. Otherwise, poor mental health and perceived negative impacts on mental health increased over the course of the three surveys and with increasing total length lockdown.

A striking finding of this study is that caregiver mental health (K6) was stable over time (except as related to total length of lockdown), while the perceived impact of the pandemic on the mental health of caregivers and children was increasingly negative. This highlights the variation in mental health experiences and measures. The K6 offers a clinical measure of mental health symptomatology, while the other is a relative measure that considers perceived impact over time. These complementary measures offer different insights into the mental health experiences of families during the pandemic. On one hand, the absolute numbers of caregivers experiencing poor mental health were similar between surveys. On the other hand, caregivers were perceiving negative impacts, which are real and concerning.

Comparable data on the mental health experiences specific to caregivers of children during the pandemic are limited. Aknin et al's review of mental health experiences during COVID-19 suggest that levels of psychological distress in general adult populations may have declined from an early peak in the first months of the pandemic.[4] However, the Pulse of the Nation (TPTN), a weekly, cross-sectional representative survey of Australian adults throughout the pandemic,[10] found that mental distress (measured with a single item that highly correlated with the K6) increased during 2020. The authors described this as driven by increasing financial stress.[10] This aligns with our finding that job or income loss during the pandemic was associated with worse mental health experiences.

In our study, length of lockdown was clearly associated with negative mental health experiences. While the RCH Poll lacked pre-pandemic data, substantially more caregivers reported poor mental health on the K6 (average 18%) in the polls throughout the pandemic than representative Australian adult data collected pre-pandemic (8% in 2017) or during the first national lockdown (11%).[11] Notably, TPTN data found that increases in mental distress were highest for parents, tripling from 8% pre-pandemic to 24% during the pandemic.[10] Our finding of increasing perceived negative impacts of the pandemic on children's mental health are consistent with the emerging reviews of the published evidence.[9]

Strengths of our study included the large cross-sectional and nationally representative surveys, which employed strong methodology (surveys piloted and included the validated K6); collected data on caregiver and child

mental health; surveyed female and male caregivers; and achieved good response rates. In other polls, indicators (frequency/prevalence) across a range of topics are almost universally consistent with more traditionally obtained estimates, providing support for the sample selection and survey administration methods. By July 2021, we found that perceived impacts of the pandemic on mental health were negative for half of children and two-thirds of adults. This highlights how general pandemic stress, and disruption to family life including employment, routines and social interactions, can adversely affect caregivers and children. Despite Australia's low incidence of COVID-19, the increases in psychological distress are consistent with data from countries with high infection rates and the increases in distress are sizeable.[4 6 8] This pattern has been described for previous health crises; following the H1N1 pandemic, a quarter of parents and 30% of children who were required to isolate met criteria for post-traumatic stress disorder (PTSD) symptoms.[12]

This study also had limitations. The reliance on caregiver report, from only one caregiver per household, means the child rating may be biased by caregiver perception. We did not collect a validated measure of children's mental health that would provide a measure of clinical impact. The finding of different social gradients for the K6 compared with the negative impact items may represent differing expectations and emphasises the need for validated measures for anticipating the longer-term consequences of lockdown, which ideally would be collected from all caregivers and children directly. The RCH Poll only samples adults who are 18 years and older, so younger caregivers are missing, and a significant proportion of caregivers did not report family income. However, the weighted and adjusted analyses support the generalisability of findings to the broader population of Australian families raising children and young people.

Australia's reliance on lockdown has meant that caregivers had to concurrently parent, work and facilitate remote schooling. Our findings show that, despite the low disease incidence, the pandemic experience has undermined caregivers' and children's mental health. Understanding specific risk factors for poorer mental health over time was beyond the scope of the cross-sectional data. However, our findings support the inequities evident in existing research. As researchers have suggested,[4 6 8] understanding and prioritising population risk groups should be the subject of ongoing research. Clearly, pandemic response and recovery planning must prioritise and incorporate mental health supports. To this end, we endorse the recommendations proposed by Samji et al.[6] These include prioritising safe access to childcare and schools; greater investment in evidence-informed mental health services including telehealth, and its integration with social care systems; and mental health promotion across communities such as work and schools. The authors also emphasise the importance of monitoring mental health. As our findings demonstrate,

mental health is changing during the pandemic. Understanding families' experiences is necessary to inform policy effort with a greater level of precision, so that we can respond to the evolving mental health needs of children and their families.

## Author affiliations
[1]Centre for Community Child Health, Murdoch Children's Research Institute, Parkville, Victoria, Australia
[2]Paediatrics, The University of Melbourne, Melbourne, Victoria, Australia
[3]Centre for Community Child Health, Royal Children's Hospital Melbourne, Parkville, Victoria, Australia
[4]General Medicine, The Royal Children's Hospital Melbourne, Parkville, Victoria, Australia
[5]Clinical Epidemiology and Biostatistics Unit, Murdoch Children's Research Institute, Parkville, Victoria, Australia

**Acknowledgements** We thank all families who took part in the Royal Children's Hospital National Child Health Polls. The National Child Health Polls are funded by The Royal Children's Hospital Foundation (Grant #n/a).

**Contributors** AMHP: Conceptualisation, validation, formal analysis, investigation, resources, data curation, writing—original draft, writing—review & editing, visualisation, supervision, project administration, funding acquisition. M-AM: Conceptualisation, methodology, validation, investigation, data curation, writing—review & editing. MH: Methodology, validation, formal analysis, investigation, data curation, writing—review & editing. AR: Conceptualisation, methodology, investigation, resources, writing—review & editing, supervision, project administration, funding acquisition. SG: Conceptualisation, methodology, investigation, resources, writing—review & editing, supervision, funding acquisition. AR and SG are joint senior authors. AP is overall guarantor.

**Funding** This work was supported by The Healthier Wealthier Families project, which was supported by the Helen Macpherson Smith Trust Impact Grant #9523 and The Corella Fund (Grant #n/a). The Murdoch Children's Research Institute (MCRI) administered research grants for the work and provided infrastructural support (as study sponsor) to its staff but played no role in the conduct or analysis of the research. Research at the MCRI is supported by the Victorian Government's Operational Infrastructure Support Program. AMHP was supported by The Erdi Foundation Child Health Equity (COVID-19) Scholarship (Grant# n/a). SG was supported by an NHMRC Practitioner Fellowship (#1155290).

**Competing interests** None declared.

**Patient and public involvement** Patients and/or the public were not involved in the design, or conduct, or reporting, or dissemination plans of this research.

**Patient consent for publication** Not applicable.

**Ethics approval** This study involves human participants and was approved by The Royal Children's Hospital Human Research Ethics Committee approved the research in February 2020, Reference #35254. Participants gave informed consent to participate in the study before taking part.

**Provenance and peer review** Not commissioned; externally peer reviewed.

**Data availability statement** Data are available on reasonable request and necessary approvals (e.g. ethical). The data underlying the results presented in the study are available from the Royal Children's Hospital Child Health Poll, please contact child.healthpoll@rch.org.au.

**ORCID iD**
Anna MH Price http://orcid.org/0000-0002-8117-8059

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
