## [Reviewer comments · BMJ Paediatrics Open]

ARTICLE DETAILS

TITLE (PROVISIONAL)	Child and caregiver mental health during 12 months of the COVID-19 pandemic in Australia
AUTHORS	Price, Anna MH Measey, Mary-Anne Hoq, Monsurul Rhodes, Anthea Goldfeld, Sharon

VERSION 1 – REVIEW

REVIEWER	Reviewer name: Dr. Peter Flom Institution and Country: Peter Flom Consulting West End Ave New York, United States Competing interests: None
REVIEW RETURNED	26-Jan-2022

GENERAL COMMENTS	I confine my remarks to statistical aspects of this paper. Unfortunately, I think the analysis is not correct. Rather than dichotomizing the dependent variables, they should be left in the original form. Then the authors can use ordinal logistic regression to do the analysis, rather than simply describing the results. (There's nothing wrong with describing the results -- although the measures should not be categorized the way they were, see below) -- but the regression would add a lot to the conclusions, letting the researchers control for other variables. I would do two ordinal logistic regressions. Table 1 - why was age categorized? This is a bad idea. See my blog post https://medium.com/@peterflom/what-happens-when-we-categorize-an-independent-variable-in-regression-77d4c5862b6c - similarly, for education, unless some categories were very low, all the categories should be kept. Also, I have seen some research that shows that mental health varies between those finishing any level vs. those stopping in the middle of any level.- also, income should not be treated this way. If you have actual numbers, use those. If you have 10 categories, use those. Dealing with ordinal IVs can be hard, but one option is optimal scaling.- and also, the same for amount of lockdown- for mental health, the main measure for analysis should be the continuous version of K6 and the 5 point version of impact on mental health. I am not sure what was done in the "supplementary analysis". It looks like the authors just pasted some computer output of some sort of generalized linear model, but these results don't seem to be used in the actual text and it's unclear what was done. Figure 1 - I would jitter the dates, to avoid overlapping confidence intervals and make it easier to read.
--

	Peter Flom
REVIEWER	Reviewer name: Gemma Sicouri Institution and Country: University of New South Wales Black Dog Institute United Kingdom of Great Britain and Northern Ireland United Kingdom of Great Britain and Northern Competing interests: None
REVIEW RETURNED	16-Feb-2022
GENERAL COMMENTS	The authors have done an excellent and thorough job responding to the reviewer's comments and adding to the manuscript so it is suitable as a long form paper. My only comment is for the authors to specify in the methods whether respondents reported on multiple children's mental health or an "index" child? I am assuming multiple children given they adjusted their model for family cluster, but it would be helpful to specify. Thank you for the opportunity to review this paper.

VERSION 1 – AUTHOR RESPONSE

Response to reviewers, BMJ Paediatrics Open (bmjpo-2021-001390)

Editorial and Reviewer Comment

Response

Editor in Chief: Prof Imti Choonara

Title add "a national cross-sectional survey"

Revised as suggested to: "Child and caregiver mental health during 12 months of the COVID-19 pandemic in Australia: findings from national repeated cross-sectional surveys"

Respond fully to the stats reviewer and the comments of the Associate Editor

Please see our detailed responses to the Associate Editor and Reviewer 1 below.

Associate Editor: Dr Patricia Lucas

Thank you for the submission of this paper. The statistical reviewer has asked for some substantial changes, and the other reviewer only minor ones. In addition to the amendments requested by reviewer 1, please attend to the following:

Throughout I found the phrase 'negative mental health impacts' unhelpful because it implies measured differences. I suggest this measure should be called 'perceived impact of covid on mental health'. The current phrase is particularly confusing where you are also reporting changes in a measure of poor mental health over time. Please make sure that it is always clear whether changes you report are in mental health or perceived impact. For example see: "negative mental

health impacts were more common for female compared with male caregivers...poor caregiver mental health (K6) was similar between female and male genders.”

Please see our detailed responses to Reviewer 1 below.

Thank you for highlighting the confusing wording. As suggested, we have clarified the manuscript by specifying whether the measures or changes reported are in mental health (Kessler-6) or perceived impact on mental health. Please note, we have used ‘perceived impact of the pandemic’ rather than ‘perceived impact of COVID-19’ to distinguish the indirect mental health impacts from direct viral impacts. We have also continued to refer broadly to the objective of the study as describing the mental health experiences of living through the pandemic.

And I would suggest that a striking finding here is the difference between perceived impact and measured changes – although the proportion perceiving a negative impact increased over time, and number reporting poor mental health was stable. I would welcome some discussion of the divergence of these two different ways of asking about mental wellbeing.

Thank you for drawing attention to this finding. We agree that it is striking and have added the following to the first paragraph of the Discussion: “Poor caregiver mental health (K6) was stable and similar between male and female genders. In contrast, the proportion of caregivers reporting a negative impact on mental health increased over time, and was more common for female caregivers, and older (compared with younger) children.”

And the second paragraph: “A striking finding of this study is that caregiver mental health (K6) was stable over time (except as related to total length of lockdown), while the perceived impact of the pandemic on the mental health of caregivers and children was increasingly negative. This highlights the variation in mental health experiences and measures. The K6 offers a clinical measure of mental health symptomatology, while the other is a relative measure that considers perceived impact over time. These complementary measures offer different insights into the mental health experiences of families during the pandemic. On one hand, the absolute numbers of caregivers experiencing poor mental health were similar between surveys. On the other hand, caregivers were perceiving negative impacts, which are real and concerning.”

Please rephrase this sentence – “Surveys are administered in sixth grade-equivalent (end of elementary school) English, anonymous, and respondents are remunerated with points exchangeable for department store gift vouchers.” I would encourage you not to worry about going over the suggested word count.

As suggested, this is rephrased to: “Surveys are administered in English, with a reading level equivalent to sixth grade (the end of elementary school). Responses are anonymous, and respondents are remunerated with points exchangeable for department store gift vouchers.”

The SEIFA quintiles suggest strong response bias towards more advantaged groups. I know that you have weighted the responses before analysis, but I still think it would be useful to comment on this.

As suggested, we have added the following to the first paragraph of the Results: “The SEIFA quintiles suggested strong response bias towards more advantaged groups. There were also some differences between surveys in characteristics such as the proportion of respondents caring for young children, caregiver gender, sole caregiving, and SEIFA; characteristics that were used to create the sample weights.

Reviewer 1: Dr. Peter Flom

I confine my remarks to statistical aspects of this paper. Unfortunately, I think the analysis is not correct. Rather than dichotomizing the dependent variables, they should be left in the original form.

We agree with the Reviewer that arbitrary categorisation of variables is not correct. However, our decisions to categorise variables were made in line with our hypotheses, the clinical intent and interpretation of the study, and the validity of the measures analysed. As such, we have retained the categorisation of the variables and offer justifications in the responses that follow, to reassure the Reviewers, Editors, and readers of the reliability of our approach. With regards to categorising the independent variables, we have added the following details to the Measures Table 1:

Child age: “Families and children were differentially affected by lockdown depending on whether children were attending early education centres (day-care) or school. This is because schools and early education centres were closed for different, and varying periods of the pandemic. Caregivers of school aged children were required to facilitate home learning, which was arguably more difficult for caregivers of young (compared with older) children. Similarly, families were required to care for young children when early education centres closed, which made it difficult to work from home and balance other responsibilities. Child age was used as a proxy for these experiences, and categorised to represent pre-school (0-4 years), primary/elementary school (5-11 years) and high school (12-17 years).”

Education: “Responses were trichotomised into three categories that meaningfully represented education as a socioeconomic measure for Australians: (1) “Year 12 or less” (response options: less than year 10, Year 10 or equivalent (e.g. school certificate), Year 12 or equivalent); (2) “vocational training certificate” (response options: trade/apprenticeship e.g. carpenter), certificate/diploma (e.g. Cert IV Childcare)); or (3) “university degree” (response options: undergraduate university degree, postgraduate university degree (e.g. Masters, Doctorate, PhD).” This categorisation represents education at the levels of school, Tafe (vocational), and university. This aligns with how education is typically considered in Australian studies and maximises the power of each grouping (given small numbers in the <Year 10 and Year 10 options), and the lack of a priori distinction in the socioeconomic experience of the two vocational categories.

Low income: “A binary variable based on current total household income before tax, categorised into 10 options ranging from “less than \$500 p/week” to “more than \$3,000 p/week”, plus “prefer not to say”. In 2021, to protect against the economic fallout of lockdown, the Australian federal government rapidly implemented a suite of short-term financial supports, which included an unemployment supplement (‘JobSeeker’) which doubled recipients’ social welfare benefits from \$550 to \$1,100 a fortnight, and a wage supplement for eligible businesses to retain their workforce (‘JobKeeper’). These social policy changes represented some of the largest (albeit temporary) in Australia’s history and were so significant that, by September 2020, levels of poverty and housing stress in Australia were substantially lower than the levels directly preceding COVID-19. To capture any relationships between income poverty and mental health, we created a binary variable summarising low income (“less than AU\$1,000 p/week” compared with more) based on Australian thresholds. n=767 caregivers preferred not to report income, so this variable should be interpreted with caution.”

Total length of lockdown: “Trichotomous variable based on total length of lockdown experienced by each state/territory. By 31 July 2021, the total length of COVID-19 lockdown was greatest for the state of Victoria (“Vic”, total 31 weeks); followed by the state of New South Wales (“NSW”,

total 15 weeks) and then all “other” states and territories (total range 10-12 weeks). As such, the following geographical categories were used as a proxy for total length of lockdown: (1) Victorian (most), (2) NSW, and (3) Other (least).”

Then the authors can use ordinal logistic regression to do the analysis, rather than simply describing the results. (There's nothing wrong with describing the results -- although the measures should not be categorized the way they were, see below) -- but the regression would add a lot to the conclusions, letting the researchers control for other variables.

We thank the Reviewer for their detailed consideration of the analyses. We understand that the Reviewer is asking us to use regression modelling to examine the relationship between characteristics and mental health measures, controlling for potential covariates. However, as described in the Introduction, the overarching aim was to describe the mental health experiences of caregivers and children during the pandemic. The study did not seek to understand the complex, aetiological pathways of mental health, which is beyond the scope of these repeated cross-sectional data. In their paper “The table 2 fallacy: presenting and interpreting confounder and modifier coefficients”, *Am J Epidemiol* (2013) 10.1093/aje/kws412, Westreich and Greenland articulate why it is misleading to control for other variables without a defined causal question. To avoid this fallacy and to achieve the objective of the paper, we retained the simple descriptive statistics for our main aim.

Following the same line of reasoning, our analysis of the impact of lockdown did make a causal inference. Thus, we used generalised linear models for the binomial family, controlling for the covariates that were anticipated to be a predictor of mental health, and available in the dataset.

We have clarified this in the Analysis section with: “As the primary aim of this study was to describe the mental health experiences of Australian families during the pandemic, we calculated simple weighted proportions of the mental health measures for each survey. These were described overall and by demographic, socioeconomic and pandemic-related financial and lockdown characteristics... To investigate the specific relationship between lockdown and mental health, we used generalised linear models for the binomial family to calculate the association between each mental health measure and time of the survey, according to state/territory grouping (as a proxy for total length of lockdown, see Table 1).”

I would do two ordinal logistic regressions.

Please see our responses to Comments 7 and 8 above. The suitability of ordinal logistic regressions depends on the outcome measures and underlying assumptions. It also requires a reference group, and this goes to the intention of the study, which is to understand the negative mental health experiences of the pandemic. Thus, for this paper, we have retained the original analytic approach.

Table 1 - why was age categorized? This is a bad idea. See my blog [post](https://medium.com/@peterflom/what-happens-when-we-categorize-an-independent-variable-in-regression-77d4c5862b6c)<https://medium.com/@peterflom/what-happens-when-we-categorize-an-independent-variable-in-regression-77d4c5862b6c>

Thank you for highlighting that our age categorisation was lacking a justification. Please see our response to Comment 7 above, which now explains the decision. Thank you for sharing the blog post. We note that the argument presented is 1) for linear regression analysis and 2) where the outcome of interest is continuous. The three outcome measures in our study are binary: the K6

cut-point has well-established and meaningful clinical validation, and the perceived impacts measure is categorical. As such, we have retained the original analytic approach.

- similarly, for education, unless some categories were very low, all the categories should be kept. Also, I have seen some research that shows that mental health varies between those finishing any level vs. those stopping in the middle of any level.

We recognise the Reviewer's comment that small gradations in important socioeconomic characteristics such as education can be associated with outcomes such as mental health. As described in our response to Comment 7 above, in our study, the trichotomisation meaningfully represents the education as a socioeconomic characteristic in Australia, e.g., education at (1) school, (2) Tafe (vocational), and (3) university. This also aligns with how education is typically considered in Australian studies and maximises the power of each grouping (given small numbers in the <Year 10 and Year 10 options), and the lack of a priori distinction in the vocational categories.

- also, income should not be treated this way. If you have actual numbers, use those. If you have 10 categories, use those. Dealing with ordinal IVs can be hard, but one option is optimal scaling.

Please see our responses to Comments 7 and 8 above, which justifies the categorisation of income and explains why we have retained our analytic approach.

- and also, the same for amount of lockdown

Please see our responses to Comments 7 above, which explains the categorisation of lockdown. The categorisation is also described in detail in the Measures Table 1, for the 'current lockdown' variable:

"All Australian states and territories experienced a first national, 10-week lockdown from 23 March to 1 June 2020. Victoria experienced a second extended and more severe, 20-week lockdown from 8 July-23 November 2020. There were several short (less than a week) lockdowns in the intervening months across states, before New South Wales experienced its second extended period of lockdown, which began incrementally on 26 June 2021. Victoria experienced its fifth (in total) lockdown from 16-27 July 2021, before an extended lockdown subsequently began a week later on 5 August 2021. Current lockdown was therefore categorised as follows.

- The national lockdown had eased for the collection of the June 2020 Poll, so no (denoted 'N' in Tables 2-3) Australians were in lockdown. Thus, the number/proportion not in lockdown represents the whole cohort (and is equal to the 'overall' numbers/proportions at top of Tables 2-3).

- For the September 2020 Poll, only residents of metropolitan Melbourne in the state of Victoria were in lockdown (denoted 'Y' for "yes") compared with all other Australians (N). This was categorised according to respondents who reported living in Victoria and living in "metropolitan" compared with "regional/rural" areas.

- By July 2021, many states/territories were going in and out of lockdown, and this question was added into the Poll and reported on directly with the question "are you currently under stay-at-home orders or restrictions due to COVID-19 (also known as 'lockdown')?", responses "yes" compared with "no".

The Reviewer's comment has highlighted that the use of 'cumulative lockdown' is confusing, as this is also a statistical terms. To avoid this confusion and increase clarity, we have replaced this with "total length of lockdown" throughout.

- for mental health, the main measure for analysis should be the continuous version of K6 and the 5 point version of impact on mental health

We hope that our above responses indicate that we understand the Reviewer's concern about dichotomising variables, including the dependent mental health measures. As discussed, the choice to analyse something as a continuous or binary outcome depends on the hypothesis. In our case, we were seeking to understand the negative mental health experiences of the pandemic, using two complementary measures: clinical mental health symptomatology (K6), and perceived impact of the pandemic on mental health.

In line with our hypothesis, we used the validated, clinical cut-point of the K6 for the Australian population to understand the proportion of caregivers experiencing poor mental health and to describe changes over time (see reference: Furukawa TA, Kessler RC, Slade T, et al. The performance of the K6 and K10 screening scales for psychological distress in the Australian National Survey of Mental Health and Well-Being. *Psychol Med* 2003;33(2):357-62. doi: 10.1017/s0033291702006700 [published Online First: 2003/03/08]).

For the perceived impact variable, the 5-point scale helps eliminate anchor bias and thus increases the sensitivity of the dichotomised variable in identifying a negative versus neutral or positive impact on mental health. Given that it is a self-report measure and newly developed, it was not intended to be used as a measure of severity. For this reason, we retained the binary analysis rather than the 5-point scale.

I am not sure what was done in the "supplementary analysis". It looks like the authors just pasted some computer output of some sort of generalized linear model, but these results don't seem to be used in the actual text and it's unclear what was done.

Figure 1 graphs the data in the supplementary tables. To clarify this for readers, we have presented the data in formal Tables, presenting estimated proportions and risk differences.

Figure 1 - I would jitter the dates, to avoid overlapping confidence intervals and make it easier to read.

We have retained the original graphs for the following reasons: (a) the revised Supplementary Tables provide the data points including the confidence interval upper and lower limits; (b) jittering the points may be misleading if readers interpret the added spaces as time-differences between states when the outcomes were measured; and (c) the Stata syntax for the marginsplot does not allow jittering.

Reviewer 2: Prof Gemma Sicouri

The authors have done an excellent and thorough job responding to the reviewer's comments and adding to the manuscript so it is suitable as a long form paper.

Thank you

My only comment is for the authors to specify in the methods whether respondents reported on multiple children's mental health or an "index" child? I am assuming multiple children given they adjusted their model for family cluster, but it would be helpful to specify.

As suggested, we have added the following to the Measures section to clarify: "Caregivers self-reported their mental health (Kessler-6), and the perceived impact of the pandemic on their own mental health, and the mental health of each child in their care (details in Table 1)."